# Leonurine inhibits cardiomyocyte pyroptosis to attenuate cardiac fibrosis via the TGF-β/Smad2 signalling pathway

Zhaoyi Li[1,2☯], Keyuan Chen[1,2☯], Yi Zhun Zhu[1,2]*

1 State Key Laboratory of Quality Research in Chinese Medicine, Faculty of Chinese Medicine, Macau University of Science and Technology, Macau, Taipa, China, 2 School of Pharmacy, Macau University of Science and Technology, Macau, Taipa, China

☯ These authors contributed equally to this work.
* yzzhu@must.edu.mo

**Data Availability Statement:** All relevant data are within the paper and its Supporting Information files.

**Funding:** This work was supported by the Macau Science and Technology Development Fund (FDCT

## Abstract

Cardiac fibrosis is a common cause of most cardiovascular diseases. Leonurine, an alkaloid from *Herba leonuri*, had been indicated to treat cardiovascular diseases due to its cardioprotective effects. Recently, pyroptosis, a programmed form of cell death that releases inflammatory factors, has been shown to play an important role in cardiovascular diseases, especially cardiac fibrosis. This study examined the novel mechanism by which leonurine protects against cardiac fibrosis. In rats with isoprenaline-induced cardiac fibrosis, leonurine inhibited the expression of proteins related to pyroptosis and improved cardiac fibrosis. *In vitro*, leonurine inhibited the expression of proteins related to pyroptosis and fibrosis. Additionally, leonurine regulated the TGF-β/Smad2 signalling pathway and inhibited pyroptosis to protect cardiomyocytes and improve cardiac fibrosis. Therefore, leonurine might improve cardiac fibrosis induced by isoprenaline by inhibiting pyroptosis via the TGF-β/Smad2 signalling pathway.

## Introduction

Cardiac fibrosis, which is also called myocardial calcification, is an irreversible process that causes a stiff heart and influences cardiac function, such as cardiac relaxation and contraction, leading to many cardiovascular diseases, such as heart failure, myocardial infarction, atherosclerosis, myocardial hypertrophy and other coronary heart diseases [1]. Certain stimulating factors, including transforming growth factor-β (TGF-β), interleukin-6 (IL-6), and interleukin-1 beta (IL-1β), trigger the signal transduction pathway to promote cardiomyocytes to differentiate into myofibroblasts, resulting in a large amount of fibrin production and extracellular matrix deposition. Eventually, this process leads to cardiac fibrosis [2, 3]. Pyroptosis is one of the causes of various cardiovascular diseases [4, 5]. Pyroptosis is a new type of programmed cell death caused by cysteinyl aspartate specific proteinase 1 (caspase 1) or caspase 11 [5]. Morphologically, pyroptosis has features of apoptosis and necrosis. Unlike apoptosis, pyroptosis is usually accompanied by the formation of pores, causing membrane rupture

0007/2019/AKP, 0021/2020/AGJ, 0011/2020/A1).
The National Natural Science Foundation of China
(Nos. 81973320).

**Competing interests:** Authors declare no conflict
of interest.

and cell death, especially the release of inflammatory factors [6–8]. The caspase 1-dependent pathway is the canonical pyroptosis pathway [9]. After inflammasome activation, activated caspase 1 and cleaves gasdermin D (GSDMD), which forms cell membrane pores that promote the maturation and release of proinflammatory cytokines (IL-1β and IL-18), ultimately leading to pyroptosis [4, 10]. According to literature, the NOD-like receptor protein 3 (NLRP3) inflammasome increases collagen synthesis and the expression of IL-1, IL-18, and caspase 1 in myocardial fibroblasts [11, 12], all of which are essential in the pathogenesis of cardiac fibrosis. In addition, Caspase 1 activation, GSDMD cleavage and cell membrane perforation promote the secretion of IL-1 and IL-18, which exacerbate myocardial ischaemia–reperfusion injury [13]. Leonurine, which is an alkaloid in *Herba leonuri*, can treat myocardial infarction [14], which is an acute cardiac disease [15], and attenuates cardiac fibrosis after myocardial infarction by inhibiting NADPH oxidase 4 [16]. However, there has been no report on leonurine as a treatment for heart failure, which is a chronic heart disease [17]. Therefore, in this study, the cardioprotective effect of leonurine against heart failure in rats was investigated in the heart failure rat model induced by isoprenaline (ISO), and the effect of leonurine on cardiac fibrosis was examined in a TGF-β-induced cardiac fibrosis model *in vitro*. We showed for the first time that leonurine could treat cardiac fibrosis by inhibiting pyroptosis.

## Materials and methods

### Materials

ISO was purchased from Sigma-Aldrich (St. Louis, Missouri, USA). CMC-Na was purchased from Macklin (Shanghai, China). Leonurine was synthesized by Professor Zhu Yi Zhun's lab at Fudan University according to the reported method [14] and was kindly provided. Lactate dehydrogenase (LDH) and creatine kinase (CK) kits were obtained from Nanjing Jiancheng Institute of Biotechnology (Nanjing, China). A detergent-compatible Bradford protein assay kit (Beyotime, Nantong, China) was used.

### Animal experiments

A total of 30 male Sprague–Dawley rats (4–6 weeks old, 220–250 g) were purchased from SPF (Beijing) Biotechnology Co., Ltd. (Beijing, China) (licence: SCXK (Beijing) 2019–0010). The rats were randomly divided into five groups, including the control group (n = 6), the model group (n = 6), and three leonurine groups (n = 6 per group) that were administered low (25 mg/kg/day), medium (50 mg/kg/day), and high (100 mg/kg/day) doses. The control group was subcutaneously injected with normal saline for 48 consecutive days. All rats, except for those in control group, were subcutaneously injected with ISO to establish a heart failure as previously reported [18]. A total of 5 mg/kg ISO was administered on the first day, and 2.5 mg/kg ISO was administered for the remaining 47 consecutive days. The rats in the three leonurine groups received low-, medium-, and high-dose leonurine orally for 48 consecutive days after ISO induction. No rats died during this experiment. At the end of the 48-day treatment period, the rats were anaesthetized with 2% pentobarbital (0.2 mL/100 g, i.p.), blood samples were collected from the abdominal aorta in vacutainers containing sodium heparin. Blood samples were immediately centrifuged at 4500 ×g for 15 min, and plasma samples were collected and stored at -80°C for later use. The rats were sacrificed by cervical dislocation, and the hearts were quickly removed and washed with ice-cold phosphate buffered saline and stored at -80°C for later use.

All animal procedures were conducted according to the Guidelines for the Care and Use of Laboratory Animals (Version 8) and were approved by the Animal Ethics Committee of the Animal Center of the State Key Laboratory of Quality Research of Traditional Chinese

Medicine, Macau University of Science and Technology. All surgeries were performed under sodium pentobarbital anaesthesia, and all efforts were made to minimize suffering.

## Haemodynamic measurement

At the end of the eight-week treatment, the rats were anaesthetized with 2% pentobarbitone (0.3 ml/200 g, i.p.). A metal signal detector was inserted into the right carotid artery to record the left ventricular systolic pressure (LVSP), left ventricular end-diastole pressure (LVEDP), the rising rate (+dP/dt max), and the falling rate (-dP/dt max) (ADI MPVS-Ultra Single Segment Foundation System, BME_036, Macau University of Science and Technology, Macau, China).

## Biochemical analysis

Plasma levels of LDH and CK were measured using detection kits (Nanjing Jiancheng Institute of Biotechnology, Nanjing, China), and plasma levels of IL-1β were measured using a detection kit (MULTI SCIENCES, Hangzhou, China) according to the manufacturer's instructions.

## Histology analysis

The heart tissues were embedded in paraffin and fixed in 4% paraformaldehyde for at least twenty-four hours. The samples were then cut into five mm pieces and set on slides according to a routine procedure. The heart sections were stained with Masson's trichrome, haematoxylin and eosin (H&E), and Sirius red (PSR), and ImageJ software was used for analysis (National Institutes of Health, Bethesda, MD, USA).

## Cell culture and treatments

The H9c2 rat embryonic ventricular myocardial cell line was purchased from the American Type Culture Collection (Manassas, VA) and cultured in Dulbecco's Modified Eagle Medium (containing 10% foetal bovine serum and 1% penicillin–streptomycin) at 37˚C in 95% air/5% $CO_2$. H9c2 cells were subjected to 20 ng/mL TGF-β for 30 minutes or 48 hours. H9c2 cells were pretreated with 20 μM leonurine for 4 hours before TGF-β stimulation.

## Western blot analysis

Protein was extracted from cultured H9c2 cells and rat heart tissue. The protein concentration was determined by a detergent-compatible Bradford protein assay kit (Beyotime, Nantong, China). Protein samples were subjected to 8% and 12% SDS–PAGE and blotted to a nitrocellulose filter membrane. The blots were blocked with 5% BSA, followed by incubation with the indicated primary antibodies, including mouse anti-β-actin, rabbit anti-Smad2, rabbit anti-phospho Smad2, rabbit anti-Gasdermin D, rabbit anti-cleaved Gasdermin D, rabbit anti-Caspase 1, rabbit anti-cleaved Caspase 1 (1:1000, Cell Signalling Technology), rabbit anti-α-SMA, mouse anti-COL3A1 and mouse anti-FN1 (1:500, Santa), overnight at 4˚C. The membranes were then washed three times with TBST and incubated with secondary antibodies, including HRP-conjugated goat anti-mouse IgG and HRP-conjugated goat anti-rabbit IgG (diluted 1:10,000, ICL Lab, USA), for 2 hours at room temperature. The membranes were washed three times with TBST, and ECL western blotting detection reagent (Millipore) was used to develop the immunoblots. Protein bands were visualized by a GE Amersham Imager 800 RGB (EG, USA).

## Immunofluorescence staining

H9c2 cells ($3 \times 10^4$ cells/well) were cultured on confocal dishes. After the different treatments, the cells were fixed in 4% paraformaldehyde for 10 minutes, followed by permeabilization with 0.3% Triton X-100 in PBS for 5 minutes. The cells were blocked with 5% BSA in PBS for 30 minutes at room temperature and incubated with Smad2 (1:1000, Cell Signalling Technology) antibodies at 4˚C for 16 hours. Fluorescein (FITC)-conjugated AffiniPure goat anti-rabbit IgG (H+L) secondary antibodies (1:200, 134325, Jackson ImmunoResearch) were added and incubated for 2 hours at room temperature. The cells were counterstained with DAPI (Beyotime, Nantong, China), and images were captured by a fluorescence microscope (IX73, Olympus).

## Measurement of ROS generation

The generation of intracellular reactive oxygen species (ROS) was examined using a Reactive Oxygen Species Assay Kit (Solarbio, Beijing, China) according to the manufacturer's instructions. Briefly, H9c2 cells ($2 \times 10^4$ cells/well) were grown in 6-well plates and subjected to different treatments. The cells were collected and centrifuged at 2000 rpm/min for 5 minutes, and the supernatant was discarded. Then, the H9c2 cells were incubated with 5 μM 2,7-dichlorofluorescein diacetate (DCFH-DA) at 37˚C for 20 minutes, washed three times with PBS, and resuspended in the flow tubes in 500 μL of PBS. After being resuspended, the cells were analysed by using a flow cytometer (BD FACSAria III, USA).

## Statistical analysis

The data are expressed as the mean ± standard error of the mean (mean ± SEM). All data analysis was performed with GraphPad Prism 8.0 software (GraphPad-Prism Software Inc., San Diego, CA, USA). Differences between the mean values of multiple groups were analysed by one-way analysis of variance with Tukey's test for post hoc comparisons. Statistical significance was considered at $P < 0.05$.

## Results

### Leonurine improved cardiac fibrosis and exerted cardioprotective effects *in vivo*

To examine the effects of leonurine on ISO-induced cardiac fibrosis, ISO-induced rats were treated with leonurine. After ISO induction, LVEDP, +dP/dt max, and -dP/dt max increased 1.62- ($P < 0.01$), 2.36- ($P < 0.01$), and 2.06-fold ($P < 0.01$), respectively, and LVSP decreased 1.75-fold ($P < 0.01$), which indicated that rat cardiac function worsened after ISO induction (Fig 1). In addition, LDH and CK increased 1.48- ($P < 0.01$) and 1.41-fold ($P < 0.01$), respectively, indicating that ISO caused myocardial tissue damage (Fig 2). High-dose (100 mg/kg/day) leonurine decreased the LVEDP, +dP/dt max, and -dP/dt max by 1.33- ($P < 0.05$), 1.62- ($P < 0.05$), and 1.81-fold ($P < 0.05$), respectively, but low-dose and middle-dose leonurine could not significantly improve the cardiac function of rats with ISO-induced heart failure. Low, medium, and high doses of leonurine increased LVSP by 1.43 ($P < 0.05$), 1.65 ($P < 0.01$), and 1.74 times ($P < 0.01$), respectively, which suggested that leonurine enhanced cardiac function in rats with ISO-induced heart failure (Fig 1). Moreover, low, medium, and high doses of leonurine decreased the levels of LDH by 0.85- ($P < 0.05$), 0.68- ($P < 0.01$) and 0.81-fold ($P < 0.05$) and CK by 0.34- ($P < 0.001$), 0.26- ($P < 0.001$) and 0.49-fold ($P < 0.001$), respectively (Fig 2). These results showed that leonurine could protect cardiac tissues from damage in rats with ISO-induced heart failure.

Cardiac fibrosis was serious after ISO induction, as evidenced by the area of cardiac fibrosis shown by H&E, Masson, and PSR staining (Fig 3), and the expression of proteins related to

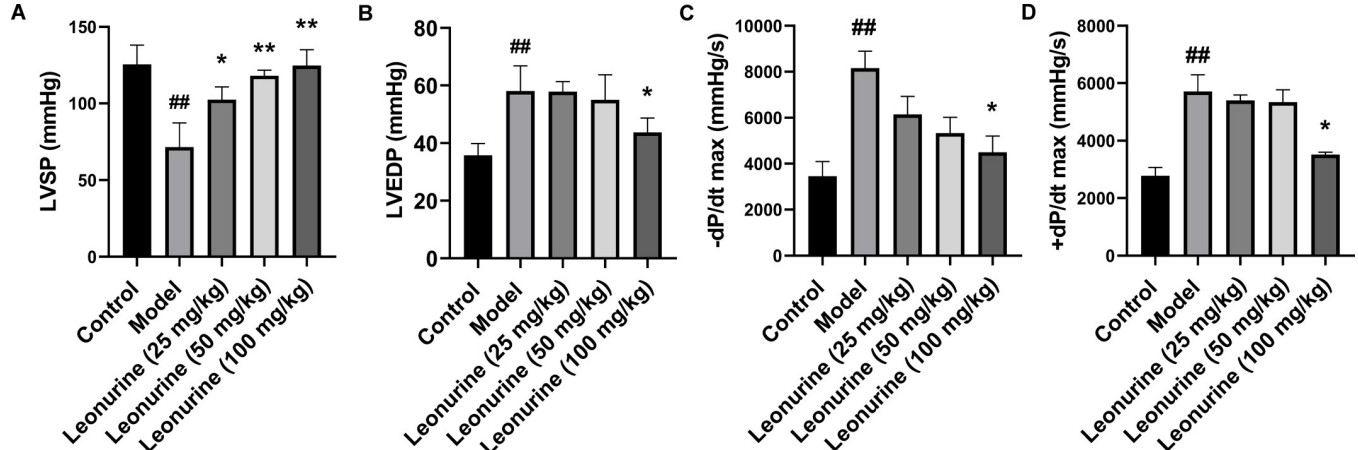

**Fig 1. Leonurine enhanced cardiac function in rats with ISO-induced heart failure by improving haemodynamic variables.** Cardiac function is responsive to haemodynamic variables, including LVSP, LVEDP, -dP/dt max, and +dP/dt max. Leonurine increased (A) LVSP level, and decreased levels of (B) LVEDP, (C) -dP/dt max, and (D) +dP/dt max compared with the model group. ##$P < 0.01$, control group vs. model group; **$P < 0.01$, *$P < 0.05$, leonurine groups vs. model group. LVSP, left ventricular systolic pressure; LVEDP, left ventricular end-diastolic pressure; -dP/dt max, the falling rate of LVEDP; +dP/dt max, the rising rate of LVEDP.

**Fig 2. Leonurine reduced myocardial injury in rats with ISO-induced heart failure.** Myocardial injury is responsive to LDH and CK. Leonurine decreased levels of (A) CK and (B) LDH in plasma in heart failure rats induced by ISO. #$P < 0.05$, ##$P < 0.01$, model group vs. control group; *$P < 0.05$, **$P < 0.01$, ***$P < 0.001$, leonurine groups vs. model group. CK, creatine kinase; LDH, lactic dehydrogenase.

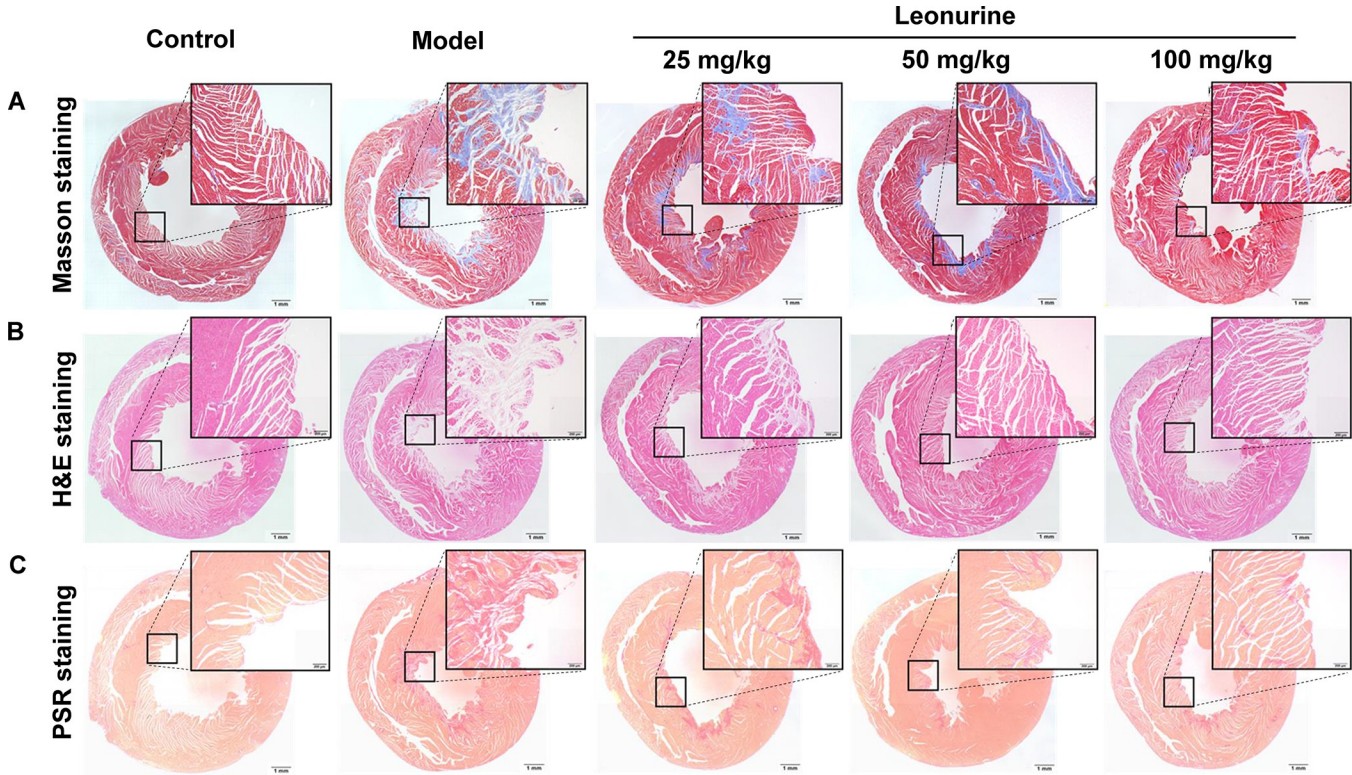

**Fig 3. Leonurine reduced the area of cardiac fibrosis in rats with ISO-induced heart failure.** (A) Masson staining on the heart, blue = fibrous collagen. (B) H&E staining on the heart, pink = fibrous collagen. (C) PSR staining on the heart, red = fibrous collagen (bar, 1 mm; magnification, ×4, ×40). H&E, haematoxylin and eosin; PSR, Sirius red.

fibrosis, including α-SMA, COL3A1 and FN1, increased 2.19- (P<0.01), 3.97- (P<0.001), and 5.78-fold (P<0.001), respectively (Fig 4). Leonurine reduced the area of cardiac fibrosis, as evidenced by the staining results (Fig 3). Furthermore, leonurine inhibited the expression of α-SMA, COL3A1, and FN1 by 2.17- (P<0.01), 1.88- (P<0.01), and 2.10-fold (P<0.0.1) in the medium-dose group, respectively, and 2.24- (P<0.01), 3.36- (P<0.01), and 2.68-fold (P<0.01) in the high-dose group, respectively (Fig 4). These results suggested that leonurine enhanced cardiac function and improved cardiac fibrosis in rats with ISO-induced heart failure.

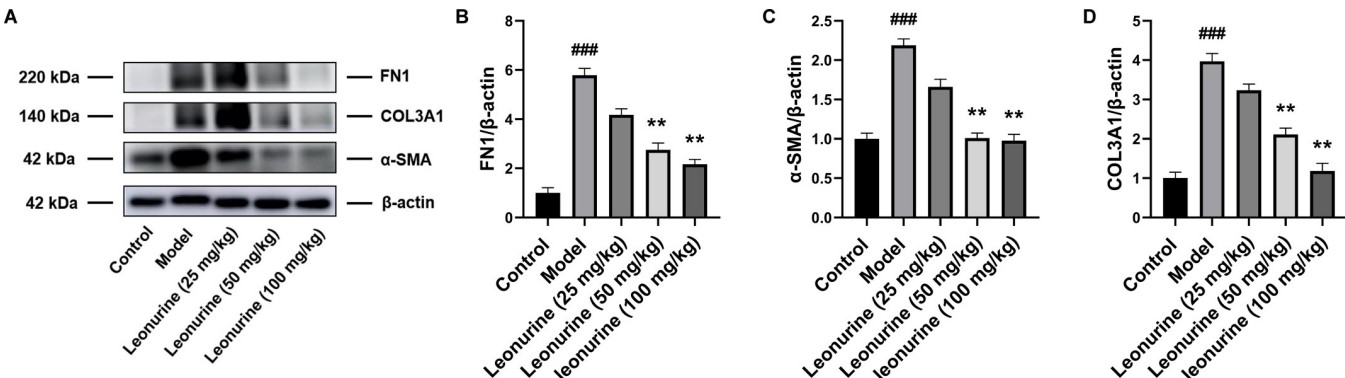

**Fig 4. Leonurine reduced the expression of proteins related to fibrosis in heart tissues from rats with ISO-induced heart failure.** (A) Compared to ISO-induced rats, leonurine decreased the expression of proteins related to fibrosis including (B) FN1, (C) α-SMA, and (D) COL3A1. ##P<0.01, ###P<0.001, model group vs. control group; **P<0.01, leonurine groups vs. model group.

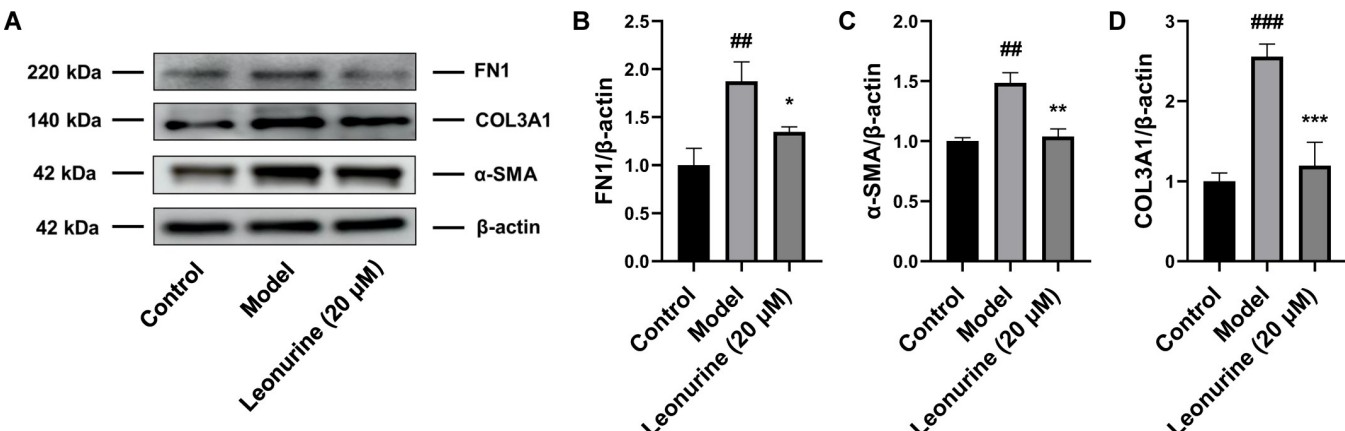

**Fig 5. Leonurine reduced the expression of proteins related to pyroptosis in heart tissues from rats with ISO-induced heart failure.** (A) Compared to ISO-induced rats, leonurine decreased expressions of pyroptosis proteins including (B) GSDMD, (C) cleaved GSDMD, (D) Caspase 1, (E) cleaved Caspase 1, and the level of (F) IL-1β in plasma. #P<0.05, ##P<0.01, ###P<0.001, model group vs. control group; *P<0.05, **P<0.01, ***P<0.001, leonurine groups vs. model group.

**Fig 6. Leonurine reduced the expression of proteins related to fibrosis in H9c2 cells stimulated by TGF-β.** (A) Compared to H9c2 cells induced by TGF-β for 48 hours, leonurine decreased the expression of proteins related to fibrosis including (B) FN1, (C) α-SMA, and (D) COL3A1. #P<0.01, ###P<0.001, model group vs. control group; *P<0.05, **P<0.01, ***P<0.001, 20 μM leonurine group vs. model group.

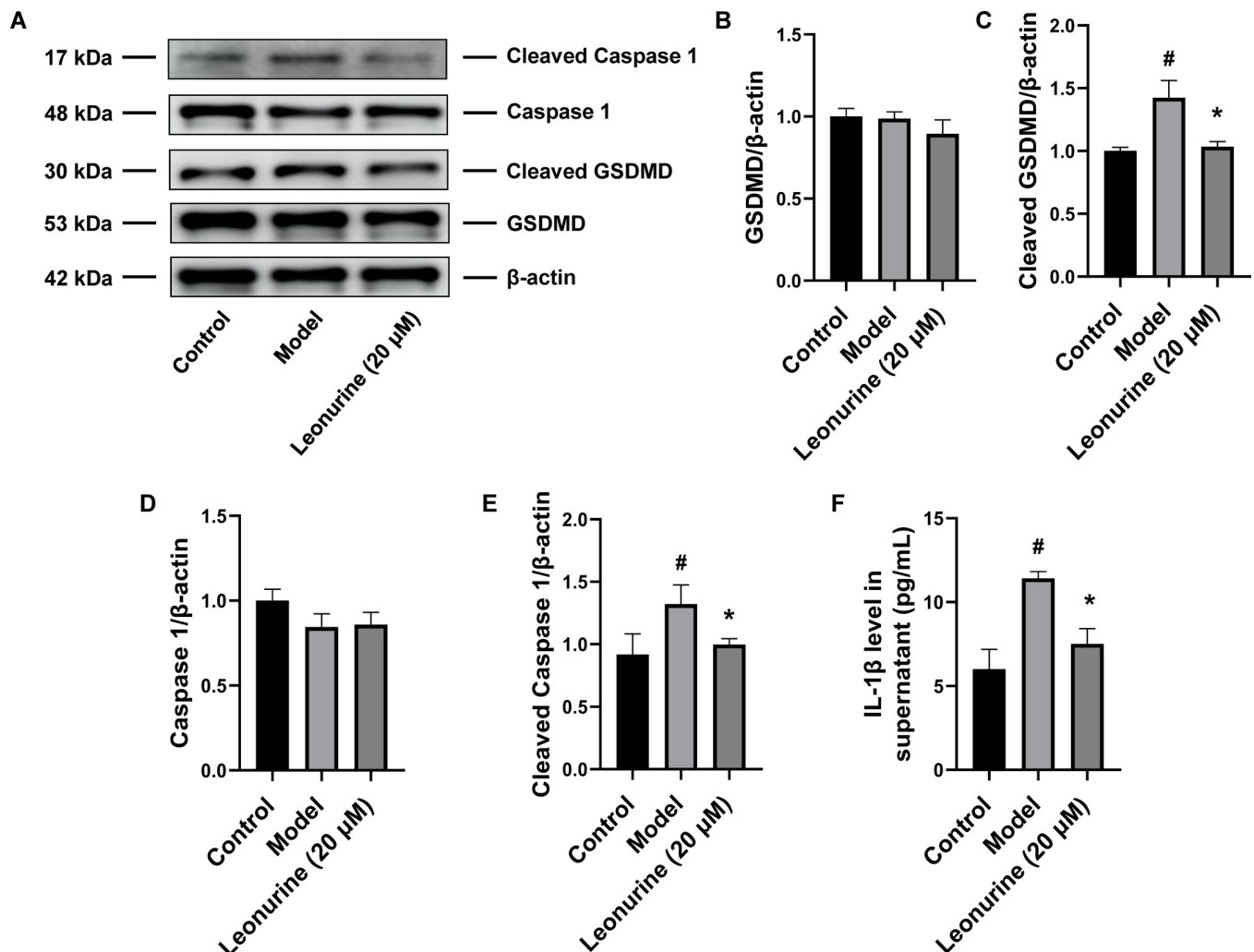

**Fig 7. Leonurine reduced the expression of proteins related to pyroptosis in H9c2 cells stimulated by TGF-β.** (A) Compared to H9c2 cells induced by TGF-β for 48 hours, leonurine decreased the expression of proteins related to pyroptosis including (B) GSDMD, (C) cleaved GSDMD, (D) Caspase 1, (E) cleaved Caspase 1, and the level of (F) IL-1β. #P<0.05, model group vs. control group; *P<0.05, 20 μM leonurine group vs. model group.

### Leonurine inhibited pyroptosis *in vivo*

Furthermore, in rats with ISO-induced heart failure, pyroptosis and inflammatory responses were exacerbated, including increases in GSDMD, cleaved GSDMD, caspase 1, cleaved caspase 1, and IL-1β by 1.56- (P<0.05), 2.87- (P<0.001), 2.14- (P<0.01), 2.67- (P<0.01), and 7.11-fold (P<0.001), respectively. However, high-dose (100 mg/kg/day) leonurine decreased the expression of proteins related to pyroptosis, including GSDMD, cleaved GSDMD, caspase 1, and cleaved caspase 1, by 1.03- (P<0.05), 1.57- (P<0.01), 1.15- (P<0.05), and 1.56-fold (P<0.01), respectively. Moreover, low, medium, and high doses of leonurine reduced the levels of IL-1β by 1.46- (P<0.05), 3.19- (P<0.001), and 4.49-fold (P<0.001), respectively, in a concentration-dependent manner, indicating that leonurine could inhibit pyroptosis after ISO induction *in vivo* (Fig 5).

### Leonurine inhibited pyroptosis *in vitro*

*In vitro*, leonurine reduced the expression of proteins related to fibrosis after TGF-β treatment, including α-SMA, COL3A1, and FN1, by 1.43- (P<0.01), 2.14- (P<0.001), and 1.39-fold

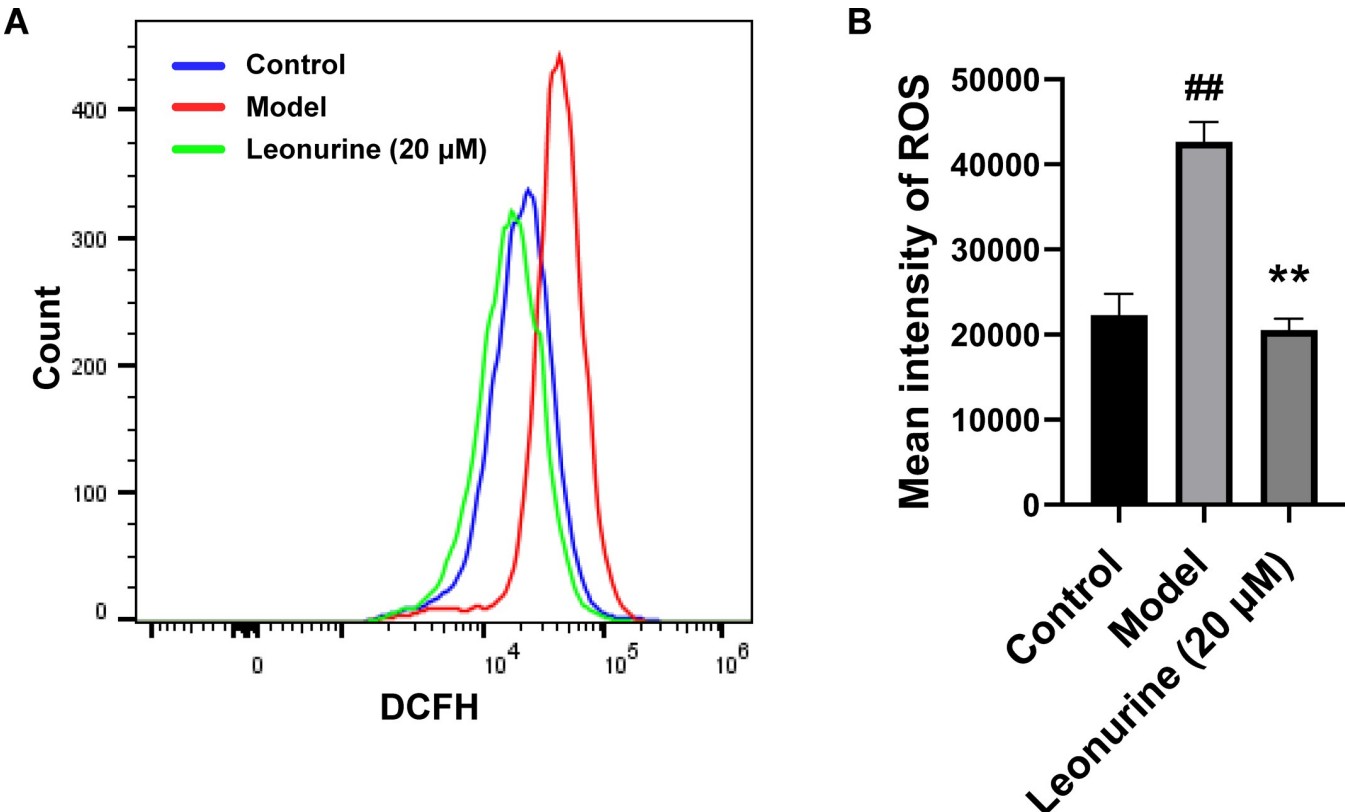

**Fig 8. Leonurine decreased ROS production by flow cytometric detection in H9c2 cells stimulated by TGF-β.** DCFH-DA was used to evaluate the intracellular ROS level. DCFH-DA is a nonfluorescent ester of the dye fluorescent product in the presence of ROS. (A) The peak value of the model group shifted to the right, and the peak value of the leonurine group shifted to the left. (B) ROS production increased with TGF-β exposure, and pretreatment with leonurine inhibited the ROS increase after TGF-β stimulation. ##P<0.01, model group vs. control group; **P<0.01, 20 μM leonurine group vs. model group. DCFH-DA, 2,7-dichlorofluorescein diacetate.

(P<0.05), respectively (Fig 6), and these results were consistent with the animal experiment results. Moreover, leonurine downregulated the expression of cleaved GSDMD and cleaved caspase 1 by 1.38- (P<0.05) and 1.38-fold (P<0.05), respectively, and decreased the level of IL-1β by 1.52-fold (P<0.05) in TGF-β-induced H9c2 cells, indicating that leonurine could inhibit pyroptosis after TGF-β stimulation *in vitro* (Fig 7). Thus, we concluded that leonurine inhibited pyroptosis to attenuate cardiac fibrosis.

### Leonurine inhibited cardiomyocyte pyroptosis via the TGF-β/Smad2 signalling pathway *in vitro*

To gain an in-depth understanding of the role of leonurine in pyroptosis during cardiac fibrosis, we analysed the effects of the TGF-β/Smad2 signalling pathway on pyroptosis induced by cardiac fibrosis. After TGF-β stimulation, the levels of ROS and phosphorylated Smad2 were increased 2.13- (P<0.01) and 6.42-fold (P<0.001), respectively, indicating that the TGF-β/Smad2 signalling pathway was activated by TGF-β (Figs 8 and 9). Leonurine reduced the levels of ROS, phosphorylated Smad2, and cleaved caspase 1 by 2.11- (P<0.01), 6.23- (P<0.001), and 1.49-fold (P<0.05), respectively (Fig 9). These results suggested that leonurine regulated the TGF-β/Smad2 signalling pathway to inhibit cardiomyocyte pyroptosis.

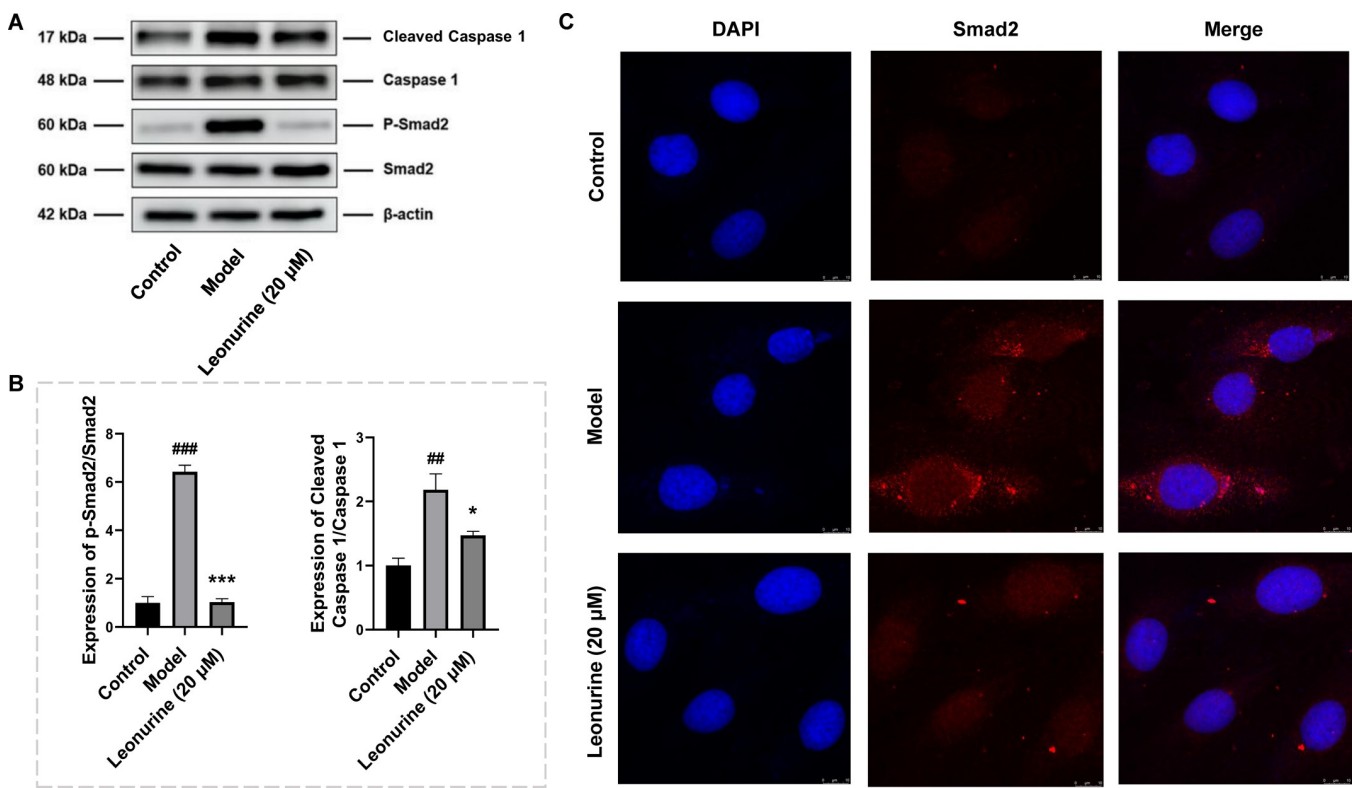

**Fig 9. Leonurine inhibited pyroptosis by regulating TGF-β/Smad2 signalling pathway in H9c2 cells stimulated by TGF-β.** (A) Compared to H9c2 cells induced by TGF-β for thirty minutes, leonurine decreased the expressions of Smad2 phosphorylation and cleaved Caspase 1 (B) The ratio of Smad2 phosphorylation to Smad2 expression and the ratio of cleaved Caspase 1 to Caspase 1 expression. (C) The fluorescence of Smad2 protein was detected under a confocal laser scanning microscope. Compare to TGF-β induced model group, the intensity fluorescence of Smad2 protein was weakened after leonurine treatment. #$P < 0.05$, ##$P < 0.01$, model group vs. control group; *$P < 0.05$, ***$P < 0.001$, 20 μM leonurine group vs. model group.

## Discussion

Heart failure, which is a global epidemic with increasing morbidity and mortality each year, has become a global public health burden [19].

A novel finding of this study was that leonurine enhanced cardiac function in rats with ISO-induced heart failure by modulating the levels of haemodynamic variables, including LVEDP, LVSP, -dP/dt max, and +dP/dt max, and decreased the levels of LDH and CK, which reduced cardiomyocyte damage and injury. Furthermore, leonurine reduced the levels of collagen and fibres to alleviate cardiac fibrosis in rats with ISO-induced heart failure. ISO, which is a catechol substance, activates the sympathetic nervous system by nonselectively exciting β receptors on cardiomyocytes, causing the production and release of catecholamines in tissue cells and increasing the concentrations of catecholamines and vasoconstrictors in the blood circulation, which increases the heart rate [20]. Furthermore, leonurine could also reduce haemodynamic parameters and the levels of LDH and CK in rats with myocardial infarction induced by coronary artery ligation [21] and heart ischaemia induced by ligation of the left coronary artery [22], thereby exerting cardioprotective effects similar to those in this study. Therefore, we hypothesize that the cardioprotective mechanism of leonurine may be independent of β-adrenergic receptors.

When β-adrenergic receptors are overactivated by ISO *in vivo*, membrane nanotubes promote inflammasome activation to spread from cardiomyocytes to cardiac fibroblasts, resulting in cardiac fibroblast pyroptosis [23]. *In vitro*, TGF-β activated the NLRP3 inflammasome in

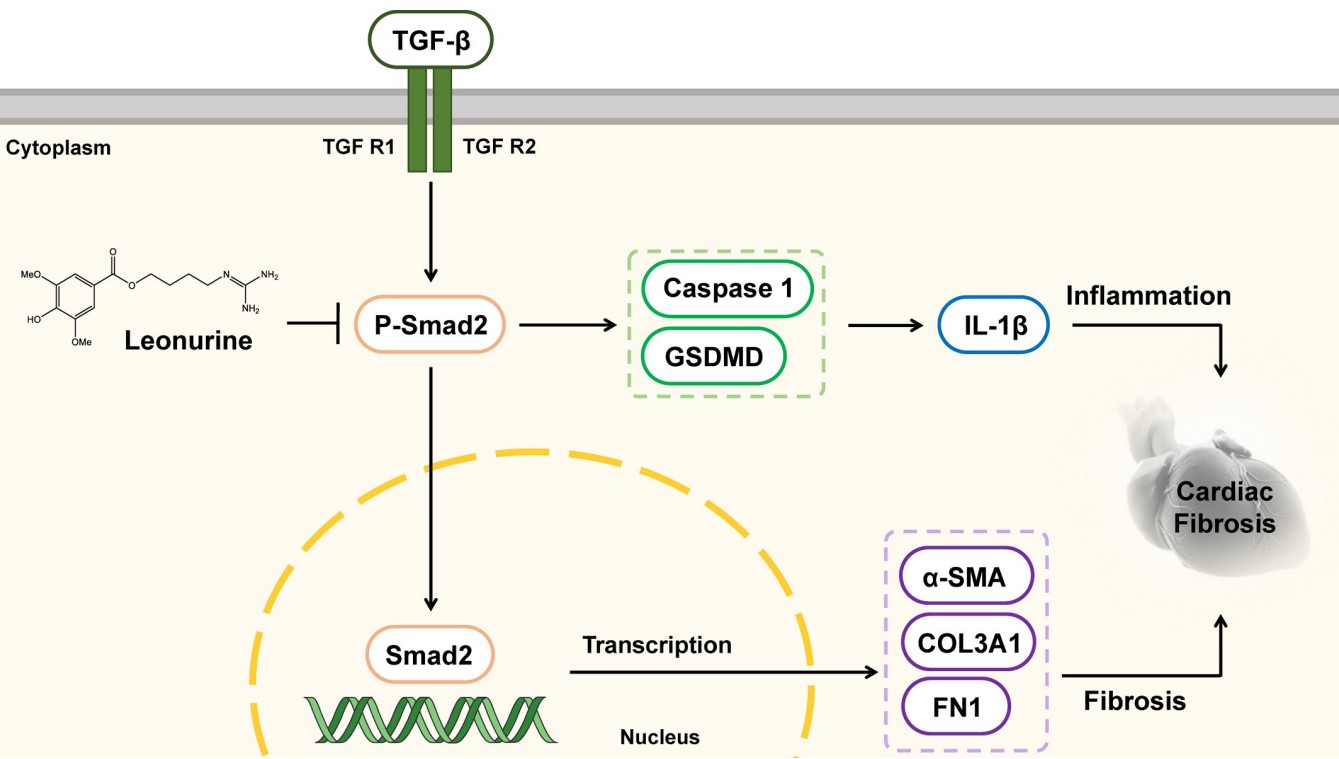

**Fig 10. Diagram of the potential mechanism of the effects of leonurine attenuating cardiac fibrosis in heart failure.** Leonurine inhibited the activation of Smad2 phosphorylation and regulated the TGF-β/Smad2 signalling pathway, thereby the expression of proteins related to pyroptosis including Caspase 1 and GSDMD. The release of a large number of inflammatory factors like IL-1β caused cardiac fibrosis and promoted the pathological process of heart failure. TGF-β, transforming growth factor-β; TGF R1, transforming growth factor receptor 1; TGF R2, transforming growth factor receptor 2; Smad2, small mothers against decapentaplegic 2; P-Smad2, phospho-Smad2; IL-1β, Interleukin 1 beta; Caspase 1, cysteinyl aspartate specific proteinase 1; GSDMD, gasdermin-D.

cardiac fibroblasts, forming a pathogenic cycle that led to the development of cardiac fibrosis [24]. Interestingly, we found that leonurine attenuated the expression of pyroptosis proteins, including Caspase 1, cleaved Caspase 1, GSDME, and cleaved GSDME, *in vivo* and *in vitro*, improving cardiac function and protecting against cardiac fibrosis, as evidenced by reductions in fibrosis areas and collagen expression. Thus, we hypothesis that pyroptosis also plays an indelible role in the development of cardiac fibrosis and that leonurine might improve cardiac fibrosis by inhibiting pyroptosis.

Reportedly, the activation of Caspase 1 promoted the release of inflammatory factors such as IL-1β [25]. Moreover, IL-1β could promote fibrosis by regulating the TGF-1β/Smad2/3 pathway [26], which is a classic pathway associated with cardiac fibrosis that plays a key role in the differentiation of cardiac fibroblasts into myocardial fibroblasts and collagen deposition [27, 28]. TGF-β1 affects R-Smad proteins, activating Smad2/3 phosphorylation, promoting profibrotic gene expression and leading to cardiac fibrosis [29]. Our study showed that leonurine could inhibit Smad2 phosphorylation in a TGF-β-stimulated cardiac fibrosis cell model, demonstrating that leonurine was involved in the regulation of the TGF-β/Smad2 signalling pathway. Furthermore, leonurine reduced ROS production in the TGF-β-stimulated cardiac fibrosis cell model. It has been reported that ROS can not only activate apoptosis [30] but also mediate fibrotic responses [31], and the results suggest that leonurine may regulate TGF-β-induced cardiac fibrosis by inhibiting apoptosis by reducing ROS production. In addition, a previous study also obtained similar results showing that leonurine could regulate the TGF-β/Smad3 signalling pathway by inhibiting Smad3 phosphorylation to treat tubulointerstitial

fibrosis in unilateral urethral obstruction (UUO) mice, and there were reductions in collagen I/III, ROS production and proinflammatory factors such as IL-1β, IL-6, and TGF-β [32]. Thus, based on these findings, we hypothesize that leonurine inhibits the activation of Caspase 1 and ROS production, preventing GSDMD cleavage and pore formation, thereby reducing the release of inflammatory factors such as IL-1β and attenuating the inflammatory response. After the activation of Caspase 1 was inhibited by leonurine, the release of inflammatory factors and Smad2 phosphorylation were reduced, which further inhibited TGF-β/Smad2 signalling pathway conduction and myofibroblast differentiation and reduced the expression of α-SMA, COL3A1, and FN1, thereby improving cardiac fibrosis.

In conclusion, our study was the first to show that leonurine enhanced cardiac function and improved cardiac fibrosis by decreasing the expression of pyroptosis proteins and fibrosis proteins *in vivo* and *in vitro*. In addition, leonurine might suppress cardiac fibrosis by inhibiting pyroptosis via the TGF-β/Smad2 signalling pathway (Fig 10). Although this study did not validate this mechanism by using inhibitors of the TGF-β/Smad2 signalling pathway or pyroptosis *in vitro*, we provide a basis for broadening the use of leonurine in the treatment of cardiac fibrosis.

## Supporting information

**S1 Table. Hemodynamic variables.**
(XLSX)

**S2 Table. OD values of CK, LDH, and IL-1β.**
(XLSX)

**S3 Table. Western blot analysis.**
(XLSX)

**S4 Table. ROS fluorescence intensity.**
(XLSX)

**S1 Raw images. Original entire blots for figures.**
(PDF)

## Author Contributions

**Writing – original draft:** Zhaoyi Li.

**Writing – review & editing:** Keyuan Chen, Yi Zhun Zhu.

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
