## [Decision Letter · Decision Letter 0]

6 Jun 2022

PONE-D-22-05865Leonurine inhibits cardiomyocyte pyroptosis against cardiac fibrosis via the TGF-β/Smad2 signaling pathwayPLOS ONE

Dear Dr. Zhu,

Thank you for submitting your manuscript to PLOS ONE. After careful consideration, we feel that it has merit but does not fully meet PLOS ONE’s publication criteria as it currently stands. Therefore, we invite you to submit a revised version of the manuscript that addresses several points raised during the review process (please, see below). Moreover, the manuscript requires significant English editing.

We look forward to receiving your revised manuscript.

Kind regards,

Luis Eduardo M Quintas, Ph.D.

Academic Editor

PLOS ONE

Journal Requirements:

Whilst you may use any professional scientific editing service of your choice, PLOS has partnered with both American Journal Experts (AJE) and Editage to provide discounted services to PLOS authors. Both organizations have experience helping authors meet PLOS guidelines and can provide language editing, translation, manuscript formatting, and figure formatting to ensure your manuscript meets our submission guidelines. To take advantage of our partnership with AJE, visit the AJE website (http://aje.com/go/plos) for a 15% discount off AJE services. To take advantage of our partnership with Editage, visit the Editage website (www.editage.com) and enter referral code PLOSEDIT for a 15% discount off Editage services.  If the PLOS editorial team finds any language issues in text that either AJE or Editage has edited, the service provider will re-edit the text for free.

A clean copy of the edited manuscript (uploaded as the new *manuscript* file).

3. To comply with PLOS ONE submissions requirements, in your Methods section, please provide additional information on the animal research and ensure you have included details on (1) methods of sacrifice, (2) methods of anesthesia and/or analgesia, and (3) efforts to alleviate suffering.

This work was supported by the Macau Science and Technology Development Fund (FDCT 0007/2019/AKP, 0021/2020/AGJ, 0011/2020/A1). The National Natural Science Foundation of China (Nos. 81973320).

Authors declare no conflict of interest.

7. PLOS requires an ORCID iD for the corresponding author in Editorial Manager on papers submitted after December 6th, 2016. Please ensure that you have an ORCID iD and that it is validated in Editorial Manager. To do this, go to ‘Update my Information’ (in the upper left-hand corner of the main menu), and click on the Fetch/Validate link next to the ORCID field. This will take you to the ORCID site and allow you to create a new iD or authenticate a pre-existing iD in Editorial Manager. Please see the following video for instructions on linking an ORCID iD to your Editorial Manager account: https://www.youtube.com/watch?v=_xcclfuvtxQ

8. Your ethics statement should only appear in the Methods section of your manuscript. If your ethics statement is written in any section besides the Methods, please delete it from any other section. 

Reviewers' comments:

Reviewer's Responses to Questions

**Comments to the Author**

1. Is the manuscript technically sound, and do the data support the conclusions?

Reviewer #1: Partly

Reviewer #2: Yes

Reviewer #3: Yes

2. Has the statistical analysis been performed appropriately and rigorously? 

Reviewer #1: Yes

Reviewer #2: Yes

Reviewer #3: Yes

3. Have the authors made all data underlying the findings in their manuscript fully available?

Reviewer #1: Yes

Reviewer #2: Yes

Reviewer #3: Yes

4. Is the manuscript presented in an intelligible fashion and written in standard English?

Reviewer #1: No

Reviewer #2: No

Reviewer #3: No

5. Review Comments to the Author

Reviewer #1: Dear author, your article is unpublished and has relevance in the field of study. However, it is necessary to pay attention to some questions that I put below for clarification:

Introduction

Page 9

“35 … After

36 respective inflammasomes activation, activated Caspase 1 and Cleaved

37 gasdermin D (GSDMD) to formed cell membrane pores that promote the

38 maturation and release of pro-inflammatory cytokines (IL-1β and IL-18),

39 ultimately leading to pyroptosis [4,10]...”

Sugested:

After

36 respectively inflammasomes activation, activated Caspase 1 and Cleaved

37 gasdermin D (GSDMD) to form cell membrane pores that promote the

38 maturation and release of pro-inflammatory cytokines (IL-1β and IL-18),

39 ultimately leading to pyroptosis [4,10].

Pags. 9 and 10

“39…Reportedly, in myocardial

40 fibroblasts, NOD-like receptor protein 40 3 (NLRP3) inflammasome

41 increased collagen synthesis and expressions of IL-1β, IL-18 and caspase

42 1 [11,12] which are playing an indispensable role in the pathogenesis of

43 cardiac fibrosis.”

Sugested:

“39…Reportedly, in myocardial

40 fibroblasts, NOD-like receptor protein 40 3 (NLRP3) inflammasome

41 increased collagen synthesis and expressions of IL-1β, IL-18 and caspase

42 1 [11,12], which plays an indispensable role in the pathogenesis of

43 cardiac fibrosis.”

Pags 9 and 10

“ 39…Reportedly, in myocardial

40 fibroblasts, NOD-like receptor protein 3 (NLRP3) inflammasome

41 increased collagen synthesis and expressions of IL-1β, IL-18 and caspase

42 1 [11,12] which are playing an indispensable role in the pathogenesis of

43 cardiac fibrosis. In additional, Caspase 1 activation, GSDMD cleavage and

44 cell membrane perforation and promoting secretion of IL-1β and IL-18,

45 which aggravated myocardial ischemia-reperfusion injury [13].

The text is confused. I suggest be more clear. Like:

“…According to reports, the NOD-like receptor protein 3 (NLRP3) inflammasome increased collagen synthesis and the expression of IL-1, IL-18, and caspase 1 in myocardial fibroblasts [11,12], all of which are essential in the pathogenesis of cardiac fibrosis. In addition, Caspase 1 activation, GSDMD cleavage and cell membrane perforation, and promoting secretion of IL-1 and IL-18, which aggravated myocardial ischemia-reperfusion injury…”

Pag. 10

“ …Therefore, present

55 study aims to examine whether leonurine improves cardiac fibrosis and its

56 possible mechanism in pyroptosis.”

Suggested:

“…Therefore, the present

55 study aims to examine whether leonurine improves cardiac fibrosis and its

56 possible mechanism in inhibition of pyroptosis.”

Materials and methods

Pag. 13

“ 117 followed by probed with the indicated primary antibodies including mouse…”

Sugested :

… followed by incubation with the indicated primary antibodies including mouse…”

Pag. 13

“120… rabbit anti-Smad2 phosphorylation

121 (1:1000, Cell Signaling),...”

Suggested:

“ 120… rabbit anti- phospho Smad2

121 (1:1000, Cell Signaling),…”

Pag. 14

“137…and incubated with Smad2 (1:1000, Cell Signaling)

138 antibodies at 4 ℃.”

For how long?

The experimental design is not entirely clear. When does Leonurine begin to be applied to the animal?

Another issue concerns the in vitro study. There is no reference to cell culture and treatment. It is necessary to describe the culture conditions, medium, culture time. And the experimental treatment design.

Results

Pag. 16

“164 After ISO treatment, rats were administered leonurine for eight weeks.

165 Levels of left ventricular end-diastolic pressure (LVEDP), its rising and

166 falling rate (±dP/dt max), lactic dehydrogenase (LDH) and creatine kinase

167 (CK) were significantly increased 1.53-, 2.01-, 2.31-, 1.48- and 1.41-fold,

168 and left ventricular systolic pressure (LVSP) was decreased 1.71-fold in

169 rats after ISO treatment (Fig 1 and 2).”

I suggest rewriting this section. Lots of information about the measured parameters. It could put the numerical data in an ordering closer to parameter groups. Ex: left ventricular end-diastolic pressure (LVEDP), its rising and falling rate (±dP/dt max) increased as shown respectively 1.53-, 2.01- and 2.31.

Pag. 17

“183 After leonurine treatment, expressions of 183 proteins related to

184 pyroptosis were decreased in the heart from rats induced by ISO (Fig 5).”

Sugested:

183... After leonurine treatment, the expression of proteins related to

According to the figures, it is clear that there are different answers in relation to the concentrations of leonurine used. These differences are not addressed during the discussion and there is no information for the reader to choose the concentration used in the cell culture treatment. The descriptor text of the results does not even indicate when there is a change in the use of animal tissue for cell culture. Clarity in writing is necessary for the understanding of the article to be total.

Figures:

Fig. 3 Areas of cardiac fibrosis in ISO-induced rats with leonurine treatment for eight weeks. Representative pictures of left ventricles from each group after H&E staining, Masson staining, and PSR staining (magnification, ×4, ×40). Leonurine decreased areas of collagen and fibers compared to the model group. (A) Collagen and fibers areas are pink after H&E staining. (B) Collagen and fibers areas are blue after Masson staining. (C) Collagen and fibers areas are yellow after PSR staining.

The description of figure A is interchanged with that of figure B.

In figure 8 there is an error in the first line of image A. It should be cleaved Capase 1.

Considerations:

I suggest that a rereading of the article be done in order to make the writing clearer.

The authors speak of the effect of leonurine in blocking the TGFbeta/SMAD-2 signaling pathway. An experiment in which leonurine was used and compared to a pathway inhibitor would be great.

Reviewer #2: Leonurine inhibits cardiomyocyte pyroptosis against cardiac fibrosis via the TGF-β/Smad2 signaling pathway

Reviewer Recommendation and Comments

I have had the pleasure of reading Li and colleagues’ manuscript. The research manuscript aimed to answer the question whether the alkaloid Leonurine can block the progression of cardiac fibrosis and elucidate the molecular mechanism involving pyroptosis. I would like to suggest an English review by a native English-speaking reviewer for improve the understanding of the reader. Moreover, I further have a few comments on specific points in other to improve the manuscript understanding.

Major Concern:

1. Title: “Leonurine inhibits cardiomyocyte pyroptosis against cardiac fibrosis via the TGF-β/Smad2 signaling pathway.” I think the word “against” leads to the reader a misunderstanding about the main view of the manuscript. I would suggest, “Leonurine inhibits cardiomyocyte pyroptosis via the TGF-β/Smad2 signaling pathway to prevent/block (the right word depends on the answer of question 4.c.) cardiac fibrosis.

2. Abstract: The citation (line 7): “The manuscript studied the novel mechanism…” please change for “This report…”.

3. Introduction: It is missing the working hypothesis of the work. It is confusing from line 51 through the end of the paragraph. Reference 16 mentioned in the manuscript is also about the attenuation of myocardial fibrosis. Moreover, I don´t understand what the authors means that it has a great cardioprotective effect. Are the authors related this sentence with cardiac function (LEVDP, LVSP, Dp/Dt) preservation? If the answer is yes, what is the new point of this manuscript? Please clarify the working hypothesis enhancing the differences from previous publications.

4. Materials and Methods:

a. Leuronine was obtained from Fudan University; to me it seems that the substance is not commercially available. Is it as extraction? The authors should provide more information.

b. The authors should provide the total n number and mention if the treatment induced any loss of the animals.

c. Treatment protocol is confusing. To all rats, ISO was injected during 48 days. When exactly Leuronine was orally administrated? During ISO administration or after? For how long, 8 days? This may change the interpretation of the data. If the Leuronine was treated after ISO administration for 8 days, the substance reverts cardiac fibrosis. This issue should be discussed.

5. Results: The quality of Fig 3 is poor. The yellow mentioned in the text can´t be seen.

6. Discussion:

a. Paragraphs 1 and 2 says the same things, please be concise.

b. ISO is a beta-adrenergic agonist. Do the ISO-induced pyroposis mediated by beta-adrenergic receptor or is a consequence of the sustained augmentation of cardiac haemodinamic that may stimulate others mediators? If the pyroposis is mediated beta-adrenergic receptor, what is the difference of the treatments with beta adrenergic antagonist (such as atenolol) and Leuronine. Are co-treatment with both drugs synergic? The authors should provide the data or discuss this issue.

c. Leuronine is an alkaloid, if ISO was administered in concomitance to Leuronine, the physically interaction of both drugs is possible? Or, is Leuronine able to bind to beta adrenergic receptor? Another possibility is ROS chelation? These points as missed in the discussion.

d. At the end of the discussion, the authors mentioned “some limitations”, the authors should cite and discuss it better.

Minor Concern:

1. Please, provide the p values of each result in order for a better conclusion of the reader.

2. Please, revise Figure legends 1, 2, 3 and 4. The letters that label’s the panels are exchanged.´

3. It is encourage that the authors draw a picture underling the proposed molecular mechanism of Leuronide.

Reviewer #3: The authors studied the novel mechanism of leonurine on cardiac fibrosis with its cardioprotective effects. They used both in vivo animal model and experiments on H9c2 cells were conducted. The manuscript is well written. The topic is interesting, and the results presented enhancing our existing knowledge. Even though the findings are important the authors must improve the manuscript. In this regard, the results section must be rewritten. It is not clear the experiments performed and if the results described were in the animal model or in the cells. In addition, the English needs to be improved.

6. PLOS authors have the option to publish the peer review history of their article (what does this mean?). If published, this will include your full peer review and any attached files.

Reviewer #1: No

Reviewer #2: No

Reviewer #3: No

---

## [Author Response · Author response to Decision Letter 0]

27 Jul 2022

Dear editor and reviewers,

Thank you for the opportunity to revise our manuscript PONE-D-22-05865 entitled “Leonurine inhibits cardiomyocyte pyroptosis against cardiac fibrosis via the TGF-β/Smad2 signaling pathway”. We are grateful for the detailed comments and suggestions provided by the reviewers, and those input are all greatly valuable and helpful for revising and improving our paper, as well as the important guiding significance to our researches.

Please kindly see below our response to the reviewers’ comments as well as our revised manuscript. We hope we have addressed the reviewers’ concerns satisfactorily and our manuscript is now fit for publication in PLOS ONE.

All best regards

Yours sincerely

Yi Zhun Zhu 

Corresponding Author

Reviewer #1: Dear author, your article is unpublished and has relevance in the field of study. However, it is necessary to pay attention to some questions that I put below for clarification:

1. Introduction

a. Page 9, line 35-39. “After respective inflammasomes activation, activated Caspase 1 and Cleaved gasdermin D (GSDMD) to formed cell membrane pores that promote the maturation and release of pro-inflammatory cytokines (IL-1β and IL-18), ultimately leading to pyroptosis [4,10]...” Suggested: “After respectively inflammasomes activation, activated Caspase 1 and Cleaved gasdermin D (GSDMD) to form cell membrane pores that promote the maturation and release of pro-inflammatory cytokines (IL-1β and IL-18), ultimately leading to pyroptosis [4,10].”

Response: Thank you for the great suggestion. We have revised the descriptions accordingly in the Introduction section following your suggestion (Page 3, lines 37-41).

b. Pages 9 and 10, line 39-43. “Reportedly, in myocardial fibroblasts, NOD-like receptor protein 40 3 (NLRP3) inflammasome increased collagen synthesis and expressions of IL-1β, IL-18 and caspase 1 [11,12] which are playing an indispensable role in the pathogenesis of cardiac fibrosis.” Suggested: “Reportedly, in myocardial fibroblasts, NOD-like receptor protein 40 3 (NLRP3) inflammasome increased collagen synthesis and expressions of IL-1β, IL-18 and caspase 1 [11,12], which plays an indispensable role in the pathogenesis of cardiac fibrosis.”

Response: Thank you for the kind suggestion. We have revised the descriptions accordingly in the Introduction section following your suggestion (Page 4, lines 41-44).

c. Pages 9 and 10, line 39-45. “Reportedly, in myocardial fibroblasts, NOD-like receptor protein 3 (NLRP3) inflammasome increased collagen synthesis and expressions of IL-1β, IL-18 and caspase 1 [11,12] which are playing an indispensable role in the pathogenesis of cardiac fibrosis. In additional, Caspase 1 activation, GSDMD cleavage and cell membrane perforation and promoting secretion of IL-1β and IL-18, which aggravated myocardial ischemia-reperfusion injury [13].” The text is confused. I suggest be more clear. Like: “According to reports, the NOD-like receptor protein 3 (NLRP3) inflammasome increased collagen synthesis and the expression of IL-1, IL-18, and caspase 1 in myocardial fibroblasts [11,12], all of which are essential in the pathogenesis of cardiac fibrosis. In addition, Caspase 1 activation, GSDMD cleavage and cell membrane perforation, and promoting secretion of IL-1 and IL-18, which aggravated myocardial ischemia-reperfusion injury…”

Response: Thank you so much for your kind reminder. We have revised the descriptions accordingly in the Introduction section following your suggestion (Page 4, lines 41-47).

d. Page 10, line 55-56. “Therefore, present study aims to examine whether leonurine improves cardiac fibrosis and its possible mechanism in pyroptosis.” Suggested: “Therefore, the present study aims to examine whether leonurine improves cardiac fibrosis and its possible mechanism in inhibition of pyroptosis.”

Response: Thank you for the kind suggestion. We have revised the descriptions accordingly in the Introduction section following your suggestion (Page 4, lines 52-57).

2. Materials and methods

a. Page 13, line 117. “followed by probed with the indicated primary antibodies including mouse…” Suggested: “followed by incubation with the indicated primary antibodies including mouse…”

Response: Thank you so much for mentioning it. We have revised the descriptions accordingly in the Materials and methods section (Page 8, lines 131-132).

b. Page 13, line 120-121. “rabbit anti-Smad2 phosphorylation 121 (1:1000, Cell Signaling), ...” Suggested: “rabbit anti- phospho Smad2 (1:1000, Cell Signaling), …”

Response: Thank you for the great suggestion. We have revised the descriptions accordingly in the Materials and methods section (Page 8, line 133).

c. Page 14, line 138. “…and incubated with Smad2 (1:1000, Cell Signaling) antibodies at 4 °C.” For how long?

Response: Thank you for the excellent question. We have added the incubation time of primary antibodies in the Materials and methods section (Page 9, line 151). 

d. The experimental design is not entirely clear. When does Leonurine begin to be applied to the animal?

Response: Thank you so much for mentioning it. We have enriched the details of animal experiments in the Materials and Methods section (Pages 5-6, lines 69-89).

e. Another issue concerns the in vitro study. There is no reference to cell culture and treatment. It is necessary to describe the culture conditions, medium, culture time. And the experimental treatment design.

Response: Thank you for your comment. We have added detailed information on in vitro study (Pages 7-8, lines 118-125).

3. Results

a. Page 16, line 164-169. “After ISO treatment, rats were administered leonurine for eight weeks. Levels of left ventricular end-diastolic pressure (LVEDP), its rising and falling rate (±dP/dt max), lactic dehydrogenase (LDH) and creatine kinase (CK) were significantly increased 1.53-, 2.01-, 2.31-, 1.48- and 1.41-fold, and left ventricular systolic pressure (LVSP) was decreased 1.71-fold in rats after ISO treatment (Fig 1 and 2).” I suggest rewriting this section. Lots of information about the measured parameters. It could put the numerical data in an ordering closer to parameter groups. Ex: left ventricular end-diastolic pressure (LVEDP), its rising and falling rate (±dP/dt max) increased as shown respectively 1.53-, 2.01- and 2.31.

Response: Thank you for your comment. We have revised the descriptions accordingly in the Results section (Pages 11-17, lines 178-322).

b. Page 17, line 183-184. “After leonurine treatment, expressions of proteins related to pyroptosis were decreased in the heart from rats induced by ISO (Fig 5).” Suggested: “After leonurine treatment, the expression of proteins related to…”

Response: Thank you for your comment. We have revised the descriptions accordingly in the Results section (Page 14, lines 245-246).

c. According to the figures, it is clear that there are different answers in relation to the concentrations of leonurine used. These differences are not addressed during the discussion and there is no information for the reader to choose the concentration used in the cell culture treatment. The descriptor text of the results does not even indicate when there is a change in the use of animal tissue for cell culture. Clarity in writing is necessary for the understanding of the article to be total.

Response: Thank you for your comment. According to your suggestion, we have supplemented the relevant information in the Discussion section (Page 18, lines 326-332) and Results section (Pages 14-17, lines 262-322).

4. Figures

a. Fig. 3 Areas of cardiac fibrosis in ISO-induced rats with leonurine treatment for eight weeks. Representative pictures of left ventricles from each group after H&E staining, Masson staining, and PSR staining (magnification, ×4, ×40). Leonurine decreased areas of collagen and fibers compared to the model group. (A) Collagen and fibers areas are pink after H&E staining. (B) Collagen and fibers areas are blue after Masson staining. (C) Collagen and fibers areas are yellow after PSR staining. The description of figure A is interchanged with that of figure B.

Response: Thank you for your comment. We have revised the figure legends of Fig 3 accordingly.

b. In figure 8 there is an error in the first line of image A. It should be cleaved Capase 1.

Response: Thank you for your comment. We have corrected these errors in Fig 9.

5. Considerations:

a. I suggest that a rereading of the article be done in order to make the writing clearer.

Response: We appreciate your suggestion. We are very sorry for the confused expressions in this article. We have reworked this article with the help of native English speakers.

b. The authors speak of the effect of leonurine in blocking the TGFbeta/SMAD-2 signaling pathway. An experiment in which leonurine was used and compared to a pathway inhibitor would be great.

Response: Thank you for your suggestion. Although the use of leonurine in the experiments and comparison with pathway inhibitors will better clarify the mechanism of leonurine in the treatment of pyroptosis, our preliminary study has obtained a novel conclusion that leonurine inhibited pyroptosis to attenuate cardiac fibrosis by regulating TGF-β/Smad2 signaling pathway by in vivo and in vitro study. According to your suggestion, we will further use the pathway inhibitors to conduct in-depth mechanism studies and added the related information as limitations in the Discussion section (Page 21, lines 392-394).

 

Reviewer #2: Leonurine inhibits cardiomyocyte pyroptosis against cardiac fibrosis via the TGF-β/Smad2 signaling pathway Reviewer 

Recommendation and Comments

I have had the pleasure of reading Li and colleagues’ manuscript. The research manuscript aimed to answer the question whether the alkaloid Leonurine can block the progression of cardiac fibrosis and elucidate the molecular mechanism involving pyroptosis. I would like to suggest an English review by a native English-speaking reviewer for improve the understanding of the reader. Moreover, I further have a few comments on specific points in other to improve the manuscript understanding.

Major Concern:

1. Title: “Leonurine inhibits cardiomyocyte pyroptosis against cardiac fibrosis via the TGF-β/Smad2 signaling pathway.” I think the word “against” leads to the reader a misunderstanding about the main view of the manuscript. I would suggest, “Leonurine inhibits cardiomyocyte pyroptosis via the TGF-β/Smad2 signaling pathway to prevent/block (the right word depends on the answer of question 4.c.) cardiac fibrosis.”

Response: Thank you for your constructive advice. According to your suggestion, we have revised the title to “Leonurine inhibits cardiomyocyte pyroptosis to attenuate cardiac fibrosis via the TGF-β/Smad2 signaling pathway.”

2. Abstract: The citation (line 7): “The manuscript studied the novel mechanism…” please change for “This report…”.

Response: Thank you for your comment. We have revised the descriptions accordingly in abstract (Page 2, lines 7-8).

3. Introduction: It is missing the working hypothesis of the work. It is confusing from line 51 through the end of the paragraph. Reference 16 mentioned in the manuscript is also about the attenuation of myocardial fibrosis. Moreover, I don´t understand what the authors means that it has a great cardioprotective effect. Are the authors related this sentence with cardiac function (LEVDP, LVSP, Dp/Dt) preservation? If the answer is yes, what is the new point of this manuscript? Please clarify the working hypothesis enhancing the differences from previous publications.

Response: Thank you for so much for your great reminder. We have added the related information in the Introduction section (Page 4, lines 50-57).

4. Materials and Methods:

a. Leuronine was obtained from Fudan University; to me it seems that the substance is not commercially available. Is it as extraction? The authors should provide more information.

Response: Thank you so much, it is a good suggestion for our study. We have added detailed information on leonurine source in the Materials and Methods section (Page 5, lines 61-63).

b. The authors should provide the total n number and mention if the treatment induced any loss of the animals.

Response: Thank you for the kind suggestion. According to your suggestion, we have enriched the details of animal experiments in the Materials and Methods section (Page 5, line 69 and Page 6, line 82).

c. Treatment protocol is confusing. To all rats, ISO was injected during 48 days. When exactly Leuronine was orally administrated? During ISO administration or after? For how long, 8 days? This may change the interpretation of the data. If the Leuronine was treated after ISO administration for 8 days, the substance reverts cardiac fibrosis. This issue should be discussed.

Response: Thank you for your comment. According to your suggestion, we have revised the descriptions of treatment protocol in the Materials and Methods section (Pages 5-6, lines 69-89).

5. Results: The quality of Fig 3 is poor. The yellow mentioned in the text can´t be seen.

Response: We appreciate your suggestion. We have revised and updated a new Fig 3.

6. Discussion:

a. Paragraphs 1 and 2 says the same things, please be concise.

Response: Thank you so much for your kind reminder. According to your suggestion, we have revised the related information in the Discussion section (Page 18, lines 326-332).

b. ISO is a beta-adrenergic agonist. Do the ISO-induced pyroposis mediated by beta-adrenergic receptor or is a consequence of the sustained augmentation of cardiac haemodinamic that may stimulate others mediators? If the pyroposis is mediated beta-adrenergic receptor, what is the difference of the treatments with beta adrenergic antagonist (such as atenolol) and Leuronine. Are co-treatment with both drugs synergic? The authors should provide the data or discuss this issue.

Response: Thank you for your professional comment. ISO is a beta-adrenergic agonist. When β-adrenergic receptors are over-activated by ISO, membrane nanotubes promote inflammasome activation to spread from cardiomyocytes to cardiac fibroblasts, resulting in cardiac fibroblast pyroptosis [1]. β-receptor blockers such as atenolol exert their cardiovascular protective effects mainly by antagonizing β-adrenergic receptors, especially β1-receptor-mediated cardiotoxicity [2]. At present, there is no literature report on the treatment of pyroptosis by leonurine, but it has been confirmed that leonurine can inhibit the activation of NLRP3 inflammasome and reduce the release of inflammatory factors [3,4], which may be the potential mechanism of leonurine in the treatment of pyroptosis. Although there are currently no reports on the combined therapy of leonurine and β-adrenergic receptor blockers, we reviewed the literature and found that Higenamine, a benzylisoquinoline alkaloid, inhibited ISO-induced cardiac fibrosis independent of β2-adrenergic receptors [5]. Therefore, we speculate that leonurine, which also belongs to the alkaloid, may not exert its therapeutic effect through β-adrenergic receptors either. Therefore, we have revised the related information in the Discussion section (Page 19, lines 344-347).

c. Leuronine is an alkaloid, if ISO was administered in concomitance to Leuronine, the physically interaction of both drugs is possible? Or, is Leuronine able to bind to beta adrenergic receptor? Another possibility is ROS chelation? These points as missed in the discussion.

Response: Thank you for your professional comment. In this study, ISO was used as an inducer to cause heart failure in rats, and then the effects and mechanism of leonurine on heart failure rats were explored. According to the chemical structure, both ISO and leonurine are rich in electron-donating solid groups such as amino groups, hydroxy, and phenolic hydroxyl, which makes the chemical combining difficult. And according to the pharmacokinetics, depending on the different administrations and metabolic rates of ISO (S.C., Tmax=14.00±5.48 min) [6] and leonurine (P.O., Tmax=51.6±15.6 min) [7], they are also difficult to ‘get adequate’ exposure to in vivo. Therefore, we speculated there might be no physical interaction between the two drugs. This study demonstrated that leonurine could improve hemodynamic parameters and the levels of LDH and CK in ISO-induced heart failure rats. According to reports, leonurine could also reduce hemodynamic parameters and the levels of LDH and CK in the myocardial infarction rats induced by coronary artery ligation [8] and the heart ischemia rats induced by ligation of the left coronary artery [9], thereby exerting cardioprotective effects similar to this study. Therefore, we speculate that the cardioprotective mechanism of leonurine may be independent of β-adrenergic receptors. We have revised the related information in the Discussion section (Page 18, lines 322-337).

d. At the end of the discussion, the authors mentioned “some limitations”, the authors should cite and discuss it better.

Response: Thank you for the great suggestion. We have added the limitations accordingly in the Discussion section (Page 18, lines 318-320).

Minor Concern:

1. Please, provide the p values of each result in order for a better conclusion of the reader.

Response: We appreciate your suggestion. We have supplemented the p value in the Results section (Pages 11-17, lines 178-322).

2. Please, revise Figure legends 1, 2, 3 and 4. The letters that label’s the panels are exchanged.

Response: Thank you so much for mentioning it. We have revised the descriptions accordingly in the figure legends of Fig 1, 2, 3, and 4.

3. It is encouraged that the authors draw a picture underling the proposed molecular mechanism of Leuronide.

Response: Thank you for your comment. According to your suggestion, we have supplemented a summary picture of the proposed molecular mechanism of leonurine as Fig 10.  

Reviewer #3: The authors studied the novel mechanism of leonurine on cardiac fibrosis with its cardioprotective effects. They used both in vivo animal model and experiments on H9c2 cells were conducted. The manuscript is well written. The topic is interesting, and the results presented enhancing our existing knowledge. Even though the findings are important the authors must improve the manuscript. In this regard, the results section must be rewritten. It is not clear the experiments performed and if the results described were in the animal model or in the cells. In addition, the English needs to be improved.

Response: Thank you for your kind suggestion. We have revised the manuscript, refined the chapters in the Results section, and modified the language of the manuscript under the guidance of native English speakers.

References

[1] Shen J, Wu J-M, Hu G-M, Li M-Z, Cong W-W, Feng Y-N, et al. Membrane nanotubes facilitate the propagation of inflammatory injury in the heart upon overactivation of the β-adrenergic receptor. Cell Death Dis. 2020;11(11):1-11. doi: 10.1038/s41419-020-03157-7.

[2] Bencivenga L, Liccardo D, Napolitano C, Visaggi L, Rengo G, Leosco D. β-Adrenergic receptor signaling and heart failure: From bench to bedside. Heart Failure Clinics. 2019;15(3):409-19. doi: 10.1016/j.hfc.2019.02.009.

[3] Rong H, Yao C. Leonurine inhibits NLRP3 inflammasome hyperactivation and regulates macrophage M1/M2 phenotype differentiation. J Pharmaceut Pract. 2021;39(2):143-7. doi: 10.12206/j.issn.1006-0111.202101003.

[4] Zhang Q, Sun Q, Tong Y, Bi X, Chen L, Lu J, et al. Leonurine attenuates cisplatin nephrotoxicity by suppressing the NLRP3 inflammasome, mitochondrial dysfunction, and endoplasmic reticulum stress. Int Urol Nephrol. 2022:1-10. doi: 10.1007/s11255-021-03093-1.

[5] Zhu J-x, Ling W, Xue C, Zhou Z, Zhang Y-s, Yan C, et al. Higenamine attenuates cardiac fibroblast abstract and fibrosis via inhibition of TGF-Β1/Smad signaling. Eur J Pharmacol. 2021;900:174013. doi: 10.1016/j.ejphar.2021.174013.

[6] Zhou J, Yin H, Ma H, Wei S, Wen E, Zhang W, et al. Efficient and selective analytical method for the quantification of a β-adrenoceptor agonist, isoproterenol, by LC–MS/MS and its application to pharmacokinetics studies. J Liq Chromatogr R T. 2017;40(13):699-705. doi: 10.1080/10826076.2017.1348952.

[7] Li B, Wu J, Li X. Simultaneous determination and pharmacokinetic study of stachydrine and leonurine in rat plasma after oral administration of Herba Leonuri extract by LC–MS/MS. J Pharmaceut Biomed. 2013;76:192-9. doi: 10.1016/j.jpba.2012.12.029.

[8] Xu L, Jiang X, Wei F, Zhu H. Leonurine protects cardiac function following acute myocardial infarction through anti‑apoptosis by the PI3K/AKT/GSK3β signaling pathway. Mol Med Rep. 2018;18(2):1582-90. doi: 10.3892/mmr.2018.9084.

[9] Liu X, Pan L, Chen P, Zhu Y. Leonurine improves ischemia-induced myocardial injury through antioxidative activity. Phytomedicine. 2010;17(10):753-9. doi: 10.1016/j.phymed.2010.01.018.

---

## [Decision Letter · Decision Letter 1]

13 Sep 2022

Leonurine inhibits cardiomyocyte pyroptosis to attenuate cardiac fibrosis via the TGF-β/Smad2 signalling pathway

PONE-D-22-05865R1

Dear Dr. Zhu,

We’re pleased to inform you that your manuscript has been judged scientifically suitable for publication and will be formally accepted for publication once it meets all outstanding technical requirements.

Kind regards,

Luis Eduardo M Quintas, Ph.D.

Academic Editor

PLOS ONE

Additional Editor Comments (optional):

Reviewers' comments:

Reviewer's Responses to Questions

**Comments to the Author**

1. If the authors have adequately addressed your comments raised in a previous round of review and you feel that this manuscript is now acceptable for publication, you may indicate that here to bypass the “Comments to the Author” section, enter your conflict of interest statement in the “Confidential to Editor” section, and submit your "Accept" recommendation.

Reviewer #2: All comments have been addressed

2. Is the manuscript technically sound, and do the data support the conclusions?

Reviewer #2: (No Response)

3. Has the statistical analysis been performed appropriately and rigorously? 

Reviewer #2: (No Response)

4. Have the authors made all data underlying the findings in their manuscript fully available?

Reviewer #2: (No Response)

5. Is the manuscript presented in an intelligible fashion and written in standard English?

Reviewer #2: (No Response)

6. Review Comments to the Author

Reviewer #2: (No Response)

7. PLOS authors have the option to publish the peer review history of their article (what does this mean?). If published, this will include your full peer review and any attached files.

Reviewer #2: No

---

## [Editor Report · Acceptance letter]

25 Oct 2022

PONE-D-22-05865R1 

Leonurine inhibits cardiomyocyte pyroptosis to attenuate cardiac fibrosis via the TGF-β/Smad2 signalling pathway 

Dear Dr. Zhu:

I'm pleased to inform you that your manuscript has been deemed suitable for publication in PLOS ONE. Congratulations! Your manuscript is now with our production department. 

Kind regards, 

on behalf of

Dr. Luis Eduardo M Quintas 

Academic Editor

PLOS ONE